# Intrahepatic Cholestasis in Pregnancy: Review of the Literature

**DOI:** 10.3390/jcm9051361

**Published:** 2020-05-06

**Authors:** Joanna Piechota, Wojciech Jelski

**Affiliations:** 12nd Department of Obstetrics and Gynecology, Medical University of Warsaw, 00-315 Warsaw, Poland; aa.nn@op.pl; 2Department of Biochemical Diagnostics, Medical University, 15-268 Bialystok, Poland

**Keywords:** cholestasis, pregnancy, bile acids

## Abstract

Intrahepatic cholestasis of pregnancy (ICP) is the most common hepatic disorder related to pregnancy in women. It usually develops within the third trimester of pregnancy and presents with pruritus as well as elevated levels of bile acid and/or alanine aminotransferase. Clinical signs quickly resolve after delivery; however, there is a high risk of the disorder recurring in subsequent pregnancies. ICP is associated with an increased risk of perinatal complications (premature birth, respiratory disorders, even stillbirth). Elevated levels of gestational hormones and genetic predispositions are important factors for the development of ICP; among the latter, mutations in hepatobiliary transport proteins (multidrug resistance protein 3-MDR3, bile salt export pump- BSEP) play a major role. Clinical and biochemical symptoms of ICP include pruritus and increased levels of total bile acids (TBA). Serum levels of TBA should be monitored in ICP patients throughout the pregnancy as concentrations above 40 μmol/L, which define that severe ICP isassociated with an increased risk of fetal complications. Therapeutic management is aimed at reducing the clinical symptoms, normalizing maternal biochemistry and preventing complications to the fetus. Pharmacological treatment of intrahepatic cholestasis of pregnancy consists of the administration of ursodeoxycholic acid to lower the levels of TBA and possibly reduce pruritus. If the treatment fails, premature delivery should be considered.

## 1. Introduction

During pregnancy, numerous physiological and anatomical changes occur in the female body so as to ensure the best possible conditions for fetal growth. All systems and organs adapt their functions to support the healthy course of pregnancy. Pregnant mothers present with an increased demand for a number of substances such as folic acid, vitamin B12, or iron [1]. An increase in the circulating blood volume as well as changes in numerous hematological and biochemical markers are also observed as altered functions of organs, which are reflected in laboratory test results. The liver is the organ thatadapts its metabolism to the changing needs of the growing fetus. Glucose metabolism is altered due to altered sensitivity to insulin and increased gluconeogenesis. Increasing insulin resistance and hormonal changes lead to changes in the maternal lipid metabolism [2]. Another function of the liver altered in the course of pregnancy is the transport of bile and the associated progressive increase in the total bile acid (TBA) levels in blood. In most pregnancies, this increase is moderate, and the TBA levels remain within the reference range. However, some mothers may experience an excessive increase in bile acid levels as a consequence of intrahepatic cholestasis of pregnancy (ICP) [3].

## 2. Epidemiology and Etiology of Intrahepatic Cholestasis of Pregnancy

Intrahepatic cholestasis of pregnancies is one of the most common liver diseases closely related to pregnancy. The disease resolves spontaneously after delivery; however, it tends to recur in a more severe form in 45–90% of subsequent pregnancies. ICP rates are different in different ethnic groups depending on the geographical region [4]. In the Polish population, the estimated incidence rate is in the range of 1–4%. A much higher rate of up to 25% is observed among Andean Natives. In Europe, North America and Australia, ICP is encountered in about 1–2% of pregnant women. The morbidity rate increases with age and the multiplicity of pregnancy [5].

The etiology of ICP is not fully explained, with attention being drawn to the contribution from genetic, hormonal, and environmental factors. Among the genetic factors, mutations of the hepatobiliary transport protein-multidrug resistance protein 3 (*MDR3*) involved in the biliary secretion of phospholipids, are assigned a major role in the pathogenesis of ICP [6]. *MDR3* mutations are observed in approximately 16% of all ICP cases; their presence is also related to the severity of the disease and TBA levels of above 40 µmol/L [7]. However, *MDR3* protein-coding disorders may occur not only in ICP but also in hereditary low phospholipid-associated cholelithiasis (LPA), as well as in drug-induced cholestasis [8,9]. Another transport protein contributing to the development of ICP is the multidrug resistance-related protein 2 (MRP2). However, the relationship between MRP2 and the occurrence of the disease was found only in a population of South American women with no relationship being observed among Caucasians [10]. Mutations may also be related to the BSEP protein-encoding gene. The substitution of the amino acid at position 444 was shown to lead to a reduction in the bile salt export pump BSEP (bile salt export pump) protein levels and consequently to drug-induced liver damage or ICP [11]. In recent extensive studies carried out in 563 women with ICP, a very important relationship was demonstrated between ICT and the mutation in the *BSEP* gene [12]. In addition to the mutations in *MDR3* and *BSEP* genes, rare mutations within the *FIC1* gene (*ATP8B1*) present within the bile duct membrane and the *FXR* gene (*NR1H4*) were also detected in Caucasian patients diagnosed with intrahepatic cholestasis of pregnancy [13,14]. Mutations in these genes may be caused by steroid hormones or their metabolites. The involvement of sex steroids in the etiopathogenesis of intrahepatic cholestasis of pregnancy was confirmed by the incidence of ICP in multiple gestations or in patients treated with oral contraceptives. Concentrations of estrogens, progesterone, and metabolites thereof increase in the course of the pregnancy to reach peak levels during the third trimester and subsequently fall after birth, thus coinciding with the natural history of ICP [15]. However, the exact pathomechanism of sex hormones’ contribution to the development of ICP has not been fully explained as of yet. Particular cholestatic effects were demonstrated for 17-β-*D*-estradiol and sulfated metabolites of progesterone [16]. Pregnant women with ICP present with significantly higher levels of progesterone sulfate and disulfate than healthy pregnant women. The sex hormones are metabolized in the liver. A number of studies confirmed the effects of gestational hormones on the metabolism of bile acids; however, many of these studies were conducted in animal (mainly murine) models for ethical reasons [17,18]. Damaging effects of glucuronates, estrogens, and progesterone on the function of hepatobiliary transport proteins involved in the excretion of bile acids into the hepatic bile ducts were observedin vitro. This may led to changes in the composition of bile, including the ratio of hydrophilic and hydrophobic bile acids being disturbed to favor hydrophobic acids. This results in the impairment of water-soluble bile acids being transported across the placenta and excreted by maternal kidneys. The maintenance of equilibrium within the maternal-fetal pool of bile acids likely plays a particularly important role in the pathogenesis of ICP. In a normal pregnancy, bile acids are transported from the fetus to the mother, whereas in a pregnancy complicated by ICP, transplacental transport occurs in the opposite direction. Therefore, levels of bile acids are increased in both the mother and the fetus. Elevated levels of TBA are associated with the induction of oxidative stress and apoptosis, thus leading to damage to liver cells and other tissues [1].

With respect to the environmental factors, attention is drawn to dietary factors, excess erucic acid from rapeseed oil, selenium deficiency, or the impact of pesticides [19].

## 3. Clinical Symptoms of ICP

The first and major symptom of intrahepatic cholestasis of pregnancy consists ofpruritus, which occurs in the second/third trimester of pregnancy (usually after 30 weeks). Most frequently, the disorder affects volar surfaces of hands and feet. However, all body parts may be affected. Pruritus tends to exacerbate during the night, possibly entailing insomnia, irritability, and even depression [20]. In the studies conducted by Glantz et al. between 1999–2002 in more than 45,000 pregnant women, the incidence of pruritus was estimated at about 2.1% of subjects, whereas cholestasis was confirmed by additional investigations in 1.5% of subjects. [21] Rare symptoms of ICP include abdominal pain, nausea, and vomiting. Mild jaundice may develop within 4 weeks of the appearance of pruritus in about 10–15% of cases. Sometimes, fatty stools due to absorption disorders, especially lipid malabsorption, are observed in ICP patients. As a consequence, shortages of fat-soluble vitamins, including vitamin K, develop, possibly leading to elongated prothrombin times and causing perinatal hemorrhages as well as bleedings into the fetal central nervous system (CNS) [22]. Spontaneous resolution of symptoms within 2–3 weeks after the delivery is characteristic ofintrahepatic cholestasis of pregnancy. Thus, the disease is a medical problem due to the risks to the fetus. The risk of bleeding into fetal CNS has already been mentioned. In addition, ICP increases the risk of premature delivery (20–60%), intrauterine asphyxia (up to 44%), meconium staining of the amniotic fluid, and fetal bradycardia. The risk of harmful effects to the fetus increases along with maternal blood TBA levels [23]. The risk of stillbirth is increased in women with ICP pregnancy and when serum bile acids concentrations are of 100 μmol/L and more. Because most women with ICP and singleton pregnancies have TBA below this concentration, they can probably be reassured that the risk of stillbirth is similar to that of pregnant women in the general population, provided repeat bile acid testing is done until delivery. Ovadia et al. found that for singleton pregnancies, the prevalence of stillbirth was three (0.13%) of 2310 intrahepatic cholestasis of pregnancy cases in women with serum total bile acids of less than 40 μmol/L versus four (0.28%) of 1412 cases with total bile acids of 40–99 μmol/L and versus 18 (3.44%) of 524 cases for bile acids of 100 μmol/L or more [24].Germain et al. observed that bile acids increase the expression and sensitivity of oxytocin receptors in uterine muscles, potentially leading to increased rates of premature deliveries in ICP patients [25].

## 4. Laboratory Diagnostics of ICP

At present, the most sensitive biochemical marker used in the diagnostics of intrahepatic cholestasis of pregnancy is the level of total bile acids, which may be the first or the only laboratory-detected symptom. In healthy pregnant women, TBA levels are slightly and insignificantly higher than in non-pregnant women. The cut-off point is defined as the total concentration of bile acids exceeding 10 micromoles/L [26]. Numerous prospective studies estimate the risk of complications to the fetal development at TBA concentrations of above 40 micromoles/L [7]. However, Sentilhes et al. described a case of fetal death at gestation week 39 at a significantly lower TBA level [27]. On the other hand, Castano et al. demonstrated that high hypercholanemia does not always lead to the development of ICP [23].

As shown by a retrospective study by Kondrackiene et al., TBA levels alone are not a sufficiently sensitive and specific marker for this disease [22]. According to the authors, concentrations of cholic acid (CA), chenodeoxycholic acid (CDCA), and the CA/CDCA ratio are better markers [28]. Determination of total bile acids alonewithout the bile acid ratiodecreases the predictive value of positive results by more than 2% [29]. The increase in total fatty acids is mainly accompanied by an increase in the activity of aminotransferases, particularly alanine aminotransferase (ALT). An about 2–15-fold increase in ALT activity, reaching as high as more than 1000 IU/L, was observed in 60–85% of patients [30]. However, the reference range for ALT in pregnant women is still a matter of dispute. Lowering the upper limit of the reference range is postulated so as to facilitate more precise identification of pregnant women with hepatic disorders, including intrahepatic cholestasis of pregnancy. Kondrackiene et al.specified the cut-off values for ALT (31 IU/L) in women with ICP [22]. While ICP patients also present with an increase in alkaline phosphatase activity (AP), its diagnostic value is low due to placental and bone production of AP. On the other hand, no elevated gamma-glutamyltransferase activity was observed in ICP. Occasionally, dyslipidemia may be observed with increased total cholesterol, low-density lipoprotein cholesterol, and apolipoprotein levels [8].

All maternal serum parameters assayed in the liver function tests return to normal shortly after delivery.

## 5. Therapy of ICP

Fetal biochemical parameters and wellbeing are monitored in the course of the disease. Patients should follow a light and low-fat diet. Biochemical parameters such as transaminases, total bile acids, andblood coagulation profile should be analyzed once a week. An abdominal ultrasound scan should be performed to examine the liver. The assessment of fetal wellbeing should consist in the monitoring of fetal movements, cardiotocography (CTG), and fetal ultrasound scans [31].

Administration of ursodeoxycholic acid (UDCA) is the treatment of choice. The drug’s mechanism of action involves the displacement of hydrophobic bile acids to ensure the protection of hepatocytic membranes; UDCA was shown to stimulate transplacental elimination of bile acids from the fetus. Ursodeoxycholic acid is administered orally at 300 mg 2–3 times per day (or 10–16 mg/kg/day). The drug is well tolerated by patients [32]. No toxic effects on the fetus were demonstrated in the course of UDCA treatment. Intrahepatic cholestasis of pregnancy may reduce the absorption of vitamin K leading to an increase in prothrombin time, which may result in postpartum bleeding; therefore, administration of vitamin K at the dose of 10 mg is considered reasonable. In ICP-complicated pregnancies, the practice at numerous centers consists of artificial induction of labor at the 37th or 38th week of gestation [33].

However, in a trial, Chappell et al. found that for women with intrahepatic cholestasis of pregnancy, ursodeoxycholic acid was not effective in reducing a composite of adverse perinatal outcomes. Although ursodeoxycholic acid appeared to be safe, it had no clinically meaningful effect on maternal itch symptoms. It did not reduce maternal bile acid concentrations, and the reduction in alanine transaminase was of uncertain clinical significance, given that alanine transaminase is not known to be associated with the risk of stillbirth or preterm labor in intrahepatic cholestasis of pregnancy. Treatment with ursodeoxycholic acid does not reduce adverse perinatal outcomes in women with intrahepatic cholestasis of pregnancy. Therefore, its routine use for this condition should be reconsidered. Ursodeoxycholic acid is recommended in six national guidelines for the management of intrahepatic cholestasis of pregnancy, principally for the improvement of maternal symptoms and biochemical test results. Surveys of practice have found that 97% of obstetricians in the UK use ursodeoxycholic acid for treating intrahepatic cholestasis of pregnancy. However, Chappell et al. suggested that the lack of in-vivo evidence of benefit should preclude further routine clinical use of ursodeoxycholic acid, even in the absence of harm, to avoid women being offered an unproven treatment [34].

Premature delivery is contemplated in cases involving no improvement in clinical parameters and simultaneous exacerbation of clinical symptoms. The risk of fetal death should be compared with the potential risk of premature delivery. A course of corticosteroids should be administered before the 34th week of gestation to reduce fetal mortality and incidence of respiratory insufficiency and intraventricular bleeding. After the 38th week, the risk of intrauterine fetal hypoxia is increased even in cases of moderate cholestasis [35].

The guidelines of the Royal College of Obstetricians and Gynaecologists include a recommendation to induce premature delivery in ICP-complicated pregnancies in patients with severe biochemical disorders at 37+0 weeks [36].

Due to the lack of recommendations based on randomized studies, the optimum management consists ofinducing the delivery at gestation week 36–37, particularly in cases of total bile acidlevels exceeding 40 mmol/L. Vaginal vs. cesarean section delivery should be considered depending on obstetric indications. The likelihood of intrahepatic cholestasis of pregnancy recurring during a subsequent pregnancy is about 60% [37].

Intrahepatic cholestasis of pregnancy is a common hepatic disorder associated with pregnancy. Genetic, hormonal, and environmental factors are taken into account in the pathogenesis of the disorder, but the mechanism responsible for the disease has not been fully elucidated. Pruritus, developing in the third trimester of pregnancy, is the most predominant clinical symptom. The resolution of symptoms after the delivery is also characteristic. ICP presents a risk to fetal development. Laboratory diagnosis is mainly based on the determination of elevated total bile acid levels in the maternal blood. The mainstay of therapeutic management consists in the reduction of maternal symptoms and possible complications to the fetus.

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
