# Peer review of "Intrahepatic Cholestasis in Pregnancy: Review of the Literature"

_jcm, 2020, doi:10.3390/jcm9051361_

Round 1

Reviewer 1 Report

It is estimated that intrahepatic pregnancy cholestasis (ICP) affects approximately 1% of women from Northern Europe. ICP usually becomes apparent in the third trimester of pregnancy and after delivery the signs and symptoms of the condition disappear. However, they can return during later pregnancies.
ICP can cause problems for the unborn baby. This condition is associated with an increased risk of premature delivery and stillbirth. Additionally, some infants have a slow heart rate and a lack of oxygen during delivery (fetal distress).

In the reviewed manuscript, the authors reviewed current knowledge about epidemiology, etiology, diagnosis and treatment of ICP based on 35 reports from the scientific literature. Such a small number of references in the case of review work may undoubtedly be a significant and major disadvantage of this manuscript, however, careful selection of literature sources ensures that the correct end result is obtained.

Author Response

On behalf of all the authors, I would like to thank the Reviewer for the positive opinion regarding our publication.

Reviewer 2 Report

The review manuscript by Piechota and Jelski about ICP is very concise, often incorrect and does not discuss the important newer literature, giving it an outdated impression.

Below four examples to illustrate this:

1) The authors’ statement on page 1 line 39 “However, some mothers may experience an excessive 39 increase in bile acid levels and consequential intrahepatic cholestasis of pregnancy (ICP)” is incorrect and likely the other way around. Cholestasis (impaired bile flow) leads to spillover of bile salts into the systemic circulation.

2) Page 2, line 64 states “rare mutations within the FIC1 gene (ATP8B1) encoding for 64 phosphatidylserine-specific phospholipase” but ATP8B1 is not a lipase, but a flippase.

3) Page 3, line 110 “The risk of harmful effects to the fetus increases along with 110 maternal blood TBA levels [23].” As well as entire section 4 Should be updated with more recent meta-analyses, including Ovadia, Lancet 2019.

4) Page 4, line 148 states that “Administration of ursodeoxycholic acid (UDCA) is the treatment of choice.” However, a recent study by Chappell et al, Lancet 2019 convincingly shows that UDCA is not effective. This should be discussed.

Author Response

I agree with comments of reviewers and I correct the manuscript

I would like to make some clarifications in response to the reviewer  remarks regarding the publication.

  1. Indeed the Reviewer is right. I changed this sentence

However, some mothers may experience excessive increases in bile levels as a consequence of  intrahepatic gestational cholestasis (ICP).

  1. Phrase at the page 2 lines 64 “encoding for phosphatidylserine-specific phospholipase “ has been removed

The function of FIC1, the ATP8B1 gene product, has not been established, nor has the mechanism by which variations in ATP8B1 result in cholestasis. It has been hypothesised that FIC1 is an aminophospholipid translocase which translocates phosphatidylserine from the outer to the inner leaflet of the canalicular plasma membrane

  1. This point according with suggestion Reviewer became broader discussion.

The risk of stillbirth is increased in women with ICP pregnancy and when serum bile acids concentrations are of 100 μmol/L and more. Because most women with ICP and singleton pregnancies have TBA below this concentration, they can probably be reassured that the risk of stillbirth is similar to that of pregnant women in the general population, provided repeat bile acid testing is done until delivery. Ovadia et al found that for singleton pregnancies, the prevalence of stillbirth was three (0.13%) of 2310 intrahepatic cholestasis of pregnancy cases in women with serum total bile acids of less than 40 μmol/L versus four (0.28%) of 1412 cases with total bile acids of 40–99 μmol/L and versus 18 (3.44%) of 524 cases for bile acids of 100 μmol/L or more [24].

  1. Ovadia, C.; Seed, P.T.; Sklavounos, A.; Geenes, V.; Di Ilio, C.; Chambers, J.; Kohari, K.; Bacq, Y.; Bozkurt, N.; Brun-Furrer, R.; Bull, L.; Estiú, M.C.; Grymowicz, M.; Gunaydin, B.; Hague, W.M.; Haslinger, C.; Hu, Y.; Kawakita, T.; Kebapcilar, A.G.; Kebapcilar, L.; Kondrackienė, J.; Koster, M.P.H.; Kowalska-Kańka, A.; Kupčinskas, L.; Lee, R,H.; Locatelli, A.; Macias, R.I.R.; Marschall, H.U.; Oudijk, M.A.; Raz, Y.; Rimon, E.; Shan, D.; Shao, Y.; Tribe, R.; Tripodi, V.; Yayla Abide, C.; Yenidede, I.; Thornton, J.G.; Chappell, L.C.; Williamson, C. Association of adverse perinatal outcomes of intrahepatic cholestasis of pregnancy with biochemical markers: results of aggregate and individual patient data meta-analyses. Lancet 2019, 393,899-909.

  1. This point according with suggestion Reviewer became broader discussion.

In trial of Chappell et al found that  women with intrahepatic cholestasis of pregnancy, ursodeoxycholic acid was not effective in reducing a composite of adverse perinatal outcomes. Although ursodeoxycholic acid appeared to be safe, it had no clinically meaningful effect on maternal itch symptoms. It did not reduce maternal bile acid concentrations, and the reduction in alanine transaminase was of uncertain clinical significance, given that alanine transaminase is not known to be associated with the risk of stillbirth or preterm labour in intrahepatic cholestasis of pregnancy. Treatment with ursodeoxycholic acid does not reduce adverse perinatal outcomes in women with intrahepatic cholestasis of pregnancy. Therefore, its routine use for this condition should be reconsidered. Ursodeoxycholic acid is recommended in six national guidelines for management of intrahepatic cholestasis of pregnancy,6 principally for the improvement of maternal symptoms and biochemical test results. Surveys of practice have found that 97% of obstetricians in the UK use ursodeoxycholic acid for treating intrahepatic cholestasis of pregnancy.However, Chappel et al. suggested that the lack of in-vivo evidence of benefit should preclude further routine clinical use of ursodeoxycholic acid, even in the absence of harm, to avoid women being offered an unproven treatment.

  1. Chappell, L.C.; Bell, J.L.; Smith, A.; Linsell, L.; Juszczak, E.; Dixon, P.H.; Chambers, J.; Hunter, R.; Dorling, J.; Williamson, C.; Thornton, J.G.; PITCHES study group. Ursodeoxycholic acid versus placebo in women with intrahepatic cholestasis of pregnancy (PITCHES): a randomised controlled trial. Lancet 2019, 394,849-860.

The numbering of the literature has been changed.

Reviewer 3 Report

This is a comprehensive and well written review on cholestasis of pregnancy.  While there is now substantial data regarding this condition, I feel this paper adds to the breadth of information regarding this condition, especially the etiology.   Authors have done a nice job summarizing the etiology, diagnosis and treatment of ICP.

Author Response

(The authors gave the same response as above.)

Round 2

Reviewer 2 Report

My comments were addressed sufficiently and have no further comments